# Practitioners’ Views on Nicotine Replacement Therapy in Pregnancy during Lapse and for Harm Reduction: A Qualitative Study

**DOI:** 10.3390/ijerph16234791

**Published:** 2019-11-29

**Authors:** Ross Thomson, Lisa McDaid, Joanne Emery, Lucy Phillips, Felix Naughton, Sue Cooper, Jane Dyas, Tim Coleman

**Affiliations:** 1Division of Primary Care, University of Nottingham, Tower Building, University Park, Nottingham NG7 2RD, UK; Lucy.Phillips1@nottingham.ac.uk (L.P.); sue.cooper@nottingham.ac.uk (S.C.); manerba.it@hotmail.co.uk (J.D.); tim.coleman@nottingham.ac.uk (T.C.); 2School of Health Sciences, University of East Anglia, Norwich NR4 7UL, UK; L.Mcdaid@uea.ac.uk (L.M.); Joanne.Emery@uea.ac.uk (J.E.); f.naughton@uea.ac.uk (F.N.)

**Keywords:** smoking cessation, pregnancy, nicotine replacement therapy, NRT, stop smoking services, harm reduction

## Abstract

Many countries, including the UK, recommend nicotine replacement therapy (NRT) for smoking cessation during pregnancy. However, adherence to NRT is generally low, smoking lapse or relapse is common and using NRT to reduce the harm from the number of cigarettes smoked is only advocated in non-pregnant smokers. Two focus groups were conducted with 13 professionals involved in antenatal stop smoking services (SSS). The data were analysed thematically. Two themes were extracted that describe health professionals’ attitudes towards using NRT either during lapses or to reduce smoking in women who cannot quit (harm reduction). These are presented around a social–ecological framework describing three hierarchical levels of influence within smoking cessation support: (1) Organizational: providing NRT during lapses could be expensive for SSS though harm reduction could result in services helping a wider range of clients. (2) Interpersonal: participants felt using NRT for harm reduction was not compatible with cessation-orientated messages practitioners conveyed to clients. (3) Individual: practitioners’ advice regarding using NRT during smoking lapses varied; many were generally uncomfortable about concurrent smoking and NRT use and had strong reservations about recommending NRT when smoking during all but the briefest lapses. Further evidence is required to guide policy and practice.

## 1. Introduction

Smoking during pregnancy is a significant, but preventable, cause of adverse prenatal outcomes [1,2,3] and a range of childhood health problems [4]. With 10.4% of women in the United Kingdom (UK) continuing to smoke throughout pregnancy [5], reducing smoking in pregnancy has been highlighted as a priority in the 2019 National Health Service (NHS) Long Term Plan which promises cessation support in pregnancy for all [6]. The UK National Institute for Health and Care Excellence (NICE) guidance recommends that all pregnant women identified as smokers should be referred to specialist stop smoking services (SSS) [7]. In England, most women who utilise SSS are offered nicotine replacement therapy (NRT) [8] which provides nicotine in a form that does not include the toxic elements found in tobacco smoke [9]. Generally, pregnant smokers’ adherence to NRT is poor [10] and may be explained, at least partially, by an increase in nicotine metabolism in pregnancy which is likely to result in them needing higher doses to prevent cravings [11]. This could contribute to lapse and relapse during quit attempts. In general, the term lapse denotes an instance of smoking or smoking episode after a prior period of abstinence. This may be an isolated event such as a single ‘puff’ or a number of cigarettes smoked which is followed by a renewal of abstinence, or it may evolve into a relapse which typically refers to regular smoking after a period of abstinence and long-term treatment failure [12].

Smoking during pregnancy fluctuates considerably during the gestational period with repeated cycles of quit attempts and relapse [13,14]. Of women setting a quit date with SSS in England, after 4 weeks, 54% are either unsuccessful in stopping smoking or are lost to follow-up [15]; many women who manage to quit for a while in pregnancy relapse back to smoking before the end of their pregnancy or shortly after childbirth [14]. Currently the National Centre for Smoking Cessation and Training (NCSCT), which is responsible for the evidence-based treatment of pregnant women who smoke and for the training of UK smoking cessation practitioners who support them, advises that all smokers including pregnant women should, on their agreed quit date, stop smoking abruptly and not smoke. This is known as the ‘Not a Puff’ rule [16]. NCSCT gives little specific advice for lapse-management, other than suggesting that practitioners ask women to reflect on why they wanted to smoke again and encourage them to re-commit to not smoking. There is no guidance on whether or how NRT should be used in lapses. Given that some studies outside of pregnancy have demonstrated that continued use of NRT during brief lapses may prevent progression to relapse and improve cessation rates [17,18], using NRT in a similar way during pregnancy has potential for improving pregnant women’s cessation rates following quit attempts.

Another potentially beneficial use of NRT in pregnancy could be that of using NRT to smoke less (i.e., for harm reduction). Smoking in pregnancy has a dose-response relationship with a number of poor pregnancy outcomes including low birth weight [19]. Harm reduction refers to strategies designed to reduce exposures to harmful chemicals within tobacco smoke by using ‘cleaner’ sources of nicotine [20]. Currently, stop smoking services in the UK [7], along with professionals from other countries such as New Zealand [21] and Australia [22] only provide pregnant smokers with a single form of support; one that advocates and aims to facilitate abstinence. The alternative is to continue to smoke, receive no support and risk damaging the unborn child. However, for women who try to stop smoking but find this particularly hard and ultimately cannot stop smoking, a second option may be to use NRT to promote harm reduction. In the UK, in addition to smoking cessation, NICE advocates NRT for harm reduction where non-pregnant smokers are not able or do not want to stop smoking [23]. This is not advocated in pregnancy, where NRT is only recommended for stopping smoking [24].

The ease or difficulty of introducing use of NRT for harm reduction in pregnancy as part of SSS practice would very likely be influenced by smoking cessation practitioners’ views. The provision of stop smoking services to pregnant women is complex and can differ across geographical regions [8,25]. Given the diversity of stop smoking services in the UK, amongst practitioners who work within these, there may be different attitudes concerning NRT use and concurrent smoking during smoking. Consequently, to understand more about how UK smoking cessation practitioners approach issues surrounding NRT use in pregnancy, we analysed data from two focus groups with professionals involved in antenatal stop smoking services. This research forms part of a larger study intended to inform the development of an intervention to improve pregnant smokers’ adherence to NRT that is acceptable to cessation practitioners and so can be used in UK NHS behavioural support for stopping smoking.

## 2. Materials and Methods

Data collected from two focus groups were used to address the primary objective. One focus group (FG1) consisting of stop smoking service leads and policy makers was conducted to explore how pregnant women’s experiences of NRT use in pregnancy could be improved. We invited these experts as we considered them best placed to provide an insight as to how services could be improved and what changes might be feasible. The results of this focus group were used to derive messages that could be delivered to pregnant women in order to address any concerns about NRT and to encourage optimal use of nicotine products. A second focus group (FG2) comprised of stop smoking practitioners and midwives with specialist smoking cessation training was held to gain feedback on these proposed messages in terms of their perceived efficacy and acceptability to practitioners. We chose to recruit practitioners for this group as we felt they were best positioned to give their opinion based on their considerable knowledge and interactions with pregnant smokers. This manuscript follows the COREQ checklist for reporting qualitative research [26].

### 2.1. Participants

We recruited for both focus groups via email invitations sent out to individuals or organisations in the smoking in pregnancy community (see Table 1). The inclusion criteria for FG1 were that participants should have extensive experience in the field of smoking cessation in pregnancy and/or be in a senior position within a relevant organisation. All participants in FG2 were responsible for supporting pregnant women to stop smoking as this was a specific inclusion criterion. All participants in both focus groups worked in England. We aimed to recruit 5–10 participants to each group, as recommended in the literature [27,28] for being appropriate to allow each participant to fully share their views and observations, whilst enabling healthy discussion.

Approval for this research was granted by East Midlands—Nottingham 1 NHS Ethics Committee (reference: 12/EM/0388).

### 2.2. Procedure

We sent all participants a participant information sheet prior to attending focus groups. Those agreeing to participate gave written informed consent on the day of the group.

The topic guide (Appendix A) used for FG1 was informed by the preliminary findings from the research team’s previous qualitative work with stop smoking practitioners and pregnant women who had been offered NRT. This structured guide was designed to answer eight specific questions concerned with NRT use. Discussion about two of these questions contributed data to analyses for this manuscript. The first asked how services currently tackle NRT use during a lapse or relapse. For the second question, participants were first presented with evidence that smoking and concurrent NRT use would likely expose pregnant women to less tobacco smoke and probably no more nicotine than just smoking on its own and then asked their views of encouraging NRT use: for the whole of pregnancy, during lapse or relapse, or for women that didn’t want to stop smoking. These questions were devised to explore how, from practitioners’ and experts’ perspectives, women’s experiences of being guided in the use of NRT can be improved in order to encourage better use of NRT in pregnancy.

The topic guide for FG2 (Appendix A) was constructed around specific messages that could be presented to pregnant women in order to promote NRT adherence. These messages were derived from systematic reviews of the literature, interviews with pregnant women, and previous focus groups with practitioners and stop smoking experts. We presented participants with 13 slides that each highlighted a particular issue associated with NRT use and adherence and an example of a message that practitioners might convey to pregnant women in support sessions with notes or prompts to aid discussion. Responses to two of these slides contributed data to the analyses. The included messages related to; ‘not stopping NRT even if you do smoke a little’, and ‘NRT and smoking will probably expose women to no more nicotine than smoking alone’. It was made clear that participants should feel free to comment or criticise. Both focus groups were audio-recorded and subsequently transcribed verbatim. Observations and field notes were not collected in either focus group as this level of detail was not required for this particular work.

### 2.3. Analysis

Data were analysed thematically [29] adopting an inductive approach where identified themes were strongly linked to the data generated within each of the subjects discussed by participants. As each focus group involved a different topic guide, FG1 and FG2 were analysed separately. This involved one researcher (RT): (a) becoming familiar with the issues and subjects raised during the focus groups using the audio recording and transcript; (b) initial coding, applying a paraphrase or label (a ‘code’) that describes what has been interpreted in the passage as important; (c) codes were then collated into potential themes organised within the specific questions or messages presented to participants; (d) themes were then reviewed in relation to the coded extracts and checked for relevance across the data set. Next a second researcher (LM) independently reviewed the output to ensure consistency in coding and analysis and assisted in the further development of themes and sub-themes. While the aims of the two focus groups were different, there was naturally considerable overlap in the subjects that were discussed. All discussion about NRT for either harm reduction or use during lapse across the two data sets were then combined. These findings were then structured using a Social-Ecological Framework (SEF) [30]. An SEF can describe a system in terms of spheres of influence that can include societal, organisational, community, interpersonal and individual levels. We used an SEF to identify which levels key findings related to and to highlight the dynamic interactions between various personal and environmental factors related to the provision of NHS smoking cessation support to pregnant women. As no themes produced in the analysis sat within the societal or community levels, we present findings within organisational, interpersonal and individual levels of influence on smoking cessation support in an approach adopted elsewhere for this population [31]. While repeat interviews or participant checking were not undertaken, findings were discussed with our Patient and Public Involvement (PPI) Advisory Panel.

## 3. Results

Seven participants were recruited for FG1. This group was facilitated by J.D., who has considerable experience in qualitative research. The group took place in September 2018 and lasted 5 h (which also included presentations and group exercises not included in this paper). Six participants were recruited for FG2 which was conducted by R.T. (PhD) in February 2019 and lasted 3 h. RT is experienced in qualitative methods and analysis and has completed formal training on designing and moderating focus groups. Both focus groups took place at the University of Nottingham, were audio-recorded and transcribed verbatim.

Findings are organised around two themes: (i) how and why practitioners address NRT use during smoking lapse, (ii) attitudes towards NRT use and concurrent smoking for harm reduction. These are presented in terms of the perspectives of participants relating to an *organisational* (i.e., the rules, regulations and cultures of stakeholder organisations), *interpersonal* (i.e., the relationship and interactions between practitioners and pregnant women) and *individual* level (i.e., practitioners’ knowledge, attitudes and beliefs), while acknowledging that those beliefs and attitudes are affected by, and can affect, other levels.

Quotes from the focus group participants will be identified in the following manner: From focus group 1 participant 1 (FG1 P1), from focus group 2 participant 1 (FG2 P1).

### 3.1. Theme 1: How and Why Practitioners Address NRT Use during Smoking Lapse

#### 3.1.1. Subtheme 1.1: Organisational

The management and supply of NRT to pregnant women during a lapse or relapse seemed to be largely determined by the organisational restraints placed on practitioners. The emphasis was on only suppling NRT to facilitate stopping smoking, which is one of the recognised performance targets of SSS:
*[If] they’re just continually smoking and obviously if they’re using their [NRT] products as well... I’d definitely reiterate ‘we cannot continue to prescribe NRT if you are smoking every day’, we’re very clear on that. If [they] want to reset a quit date then we will start afresh... [otherwise] we do what I call ‘the walkaway’*.(FG1 P1)

Views on the supply of NRT were also influenced by financial implications, such as the cost of providing this. There was a feeling that resource limitations precluded giving women NRT to use during a lapse:
*The reason why we’re asking them to wear the patch is to replace the body with the nicotine they would have got from the cigarettes, if they’re just giving themselves nicotine from the cigarettes, they don’t need to be wearing the patch and it’s just a waste*.(FG2 P1)

#### 3.1.2. Subtheme 1.2: Interpersonal

While lapses were not seen as desirable during a quit attempt, some FG participants believed these events provided opportunities for practitioners to engage with clients and explore any issues with using NRT:
*I think it’s really important to work out what role NRT has been playing in their life over the last couple of days or weeks to then work out what the next step might be in terms of restarting a quit attempt*.(FG1 P6)

Practitioners were eager to discover how much NRT women used before a lapse and to re-emphasise how much NRT they probably need to use to become abstinent again and to remain so for the future. They believed these events could provide learning opportunities to assess whether NRT products or dosage needed changing. It was also felt that it was also important to assess women’s motivation, why they felt the need to smoke and provide encouragement to remain smoke free and work towards a successful quit.

#### 3.1.3. Subtheme 1.3: Individual

There were differences between practitioners in relation to what they considered to be a lapse or relapse. Some practitioners considered the ‘odd cigarette’ to be a *‘blip’* (FG2 P2) or *‘wobble’* (FG2 P4) and describe a lapse as regular daily smoking of a small number of cigarettes. Confusingly other practitioners referred to the ‘odd cigarette’ as a lapse and regular daily smoking of a small number of cigarettes as a relapse, while others saw relapse in terms of women abandoning their quit and
*‘going back to not using any NRT and going back to smoking’*.(FG2 P3)

At an individual level there also seemed to be differences in the advice that practitioners gave to pregnant women who reported smoking of a small number of cigarettes with some recommending time-limited continuation of NRT:
*… ‘Well carry on wearing the patches, let’s make sure we’re back on track’, but I’m not going to let that go on for another visit*.(FG1 P1)

Other practitioners, however, when faced with women who returned to smoking a small amount (i.e., one or two), on a number of consecutive days, would advocate stopping NRT and a temporary return to smoking before recommitting to quitting:
I said ‘that’s fine, OK, just get rid of the cigarettes you’ve got, smoke them. Don’t wear your patches, don’t do anything’—this is my advice—and then we’ll start again’ because otherwise it’s too blurry isn’t it?(FG2 P6)

Practitioners generally felt that proactively advising women about how they should use NRT during smoking lapses was not something that should be done in initial consultations. There was concern that any messages relating that NRT could be used during brief lapses might be misinterpreted as giving women permission to use NRT and smoke whilst not trying to stop smoking. For this reason, practitioners generally dealt with lapses as and when each incident arose, relying on women to tell them about these.

### 3.2. Theme 2: Attitudes towards NRT Use and Concurrent Smoking for Harm Reduction

#### 3.2.1. Subtheme 2.1: Organisational

Advocating that pregnant women could use NRT to cut down on smoking (i.e., harm reduction), rather than just for cessation, was seen by some participants to have a potential advantage in terms of engaging with women at an organisational level:
*I’m thinking that actually this would be a way that would really increase the number of women that we’re getting through the door*.[i.e., for support with smoking in pregnancy] (FG1 P7)

It was felt that currently only women who were motivated to try to stop smoking were engaging with services and women who were less confident or not ready to stop were not. Enabling women who were not ready to stop to receive education and support without the pressure of having to stop smoking straightaway was thought by some participants as a way of getting more women to question their smoking and perhaps encourage more into quitting.

However, other participants thought adopting such an approach might condone smoking and was not consistent with their services’ philosophy; for this to occur, changes such as re-naming SSS might be required:
We call ourselves ‘a stop smoking service’... [smoking] is not an option, maybe that isn’t something that the stop smoking service does, maybe that’s an option for a different part of the system?(FG1 P6)

#### 3.2.2. Subtheme 2.2: Interpersonal

Participants were particularly concerned that supporting or encouraging pregnant women to continue to smoke and take NRT was not compatible with the key messages they were trying to convey during their consultations:
*And if you go back to why we’re trying to get them to give up—these are pregnant women, these aren’t just normal smokers—there is no safe level of smoking for these ladies so it contradicts the other messages that we’re trying to pass on*.(FG2 P1)

There was a strong feeling that women should be encouraged to adhere to the ‘Not a Puff’ message and that one of the things they try to communicate to women is that the only way to avoid risks is to completely stop smoking. Practitioners felt that if women were given the choice to stop smoking with NRT or continue smoking with NRT that they would pick the perceived to be the easier option, especially if they are not fully committed to quitting:
The thing they’d really want to hear is ‘I can use a patch and look as if I’m making an effort but I’m allowed to still keep smoking, yey!. So will we ever get any quitters?(FG1 P1)

#### 3.2.3. Subtheme 2.3: Individual

Participants were asked for their opinions on whether giving pregnant women NRT to use while they continued to smoke to promote harm reduction, defined as the reduction of exposure to carbon monoxide and other harmful components of cigarette smoke, was acceptable. The experts/service leads in FG1 predominately felt they would need more evidence before they would be convinced that this was a credible option:
*I think if there’s evidence that harm reduction is better than not giving it a go at all then why not*.(FG1 P4)

There was also a feeling that there would need to be a long-term plan in place that would encourage women to cut-down their smoking with a view to eventually quitting.

When practitioners were asked what they thought about messages that explained that smoking and using NRT would expose women to less tobacco smoke and probably no more nicotine than just smoking on its own, this elicited a strong negative reaction as it went against both their personal beliefs and training:
*Because what I’ve always believed and thought was that when people reduce, they compensatory smoke, so they inhale deeper and certainly what we’ve found is carbon monoxide levels go up*.(FG2 P3)

Practitioners indicated that this was not a message that they felt comfortable delivering to pregnant women because as far as they were concerned it was incompatible with what they were trying to achieve and that was a smoke free pregnancy and healthy baby:
*I would not be discussing this with them because it is almost giving them the message that it’s OK to carry on smoking*.(FG2 P1)

There was, however, a feeling that this message might be reassuring for the practitioner to know:
*I think this [message]—is good for us to know but we really don’t need to be telling anybody else about it*.(FG2 P4)

## 4. Discussion

This study reports the beliefs and attitudes of stop smoking professionals on the use of NRT and concurrent smoking in pregnant smokers in relation to lapse or harm reduction. It seems that smoking cessation practitioners are reluctant to advise NRT use and concurrent smoking for anything but the briefest of smoking lapses, and that using NRT in pregnancy to promote harm reduction was not an approach practitioners felt they could advocate.

In this study we found that some practitioners advise women who smoke a small amount (i.e., one or two cigarettes), on a number of consecutive days after their quit date to stop using NRT and prepare for another quit attempt. While these behaviours may be influenced by organisational level approaches, it may also be due to the lack of specific advice and training for practitioners on this issue, which in turn, may be the result of insufficient evidence concerning this issue in pregnancy. That some practitioners advocate the cessation of NRT when any smoking occurs is strikingly different from what clinicians recommend for those on agonist therapies for other drug dependencies when lapses occur; for example, the advice is that pregnant, opioid-dependent women who lapse may need their agonist treatment increased, not stopped, to maintain therapeutic plasma levels, and prevent further relapse [32].

The case for the continuation of NRT during a lapse is supported by studies of non-pregnant people using NRT. These studies reported that sustained use of an NRT patch after a lapse decreased the proportion of participants transitioning to relapse [18,33,34]. It is suggested that NRT use during a lapse episode could increase long-term abstinence via several mechanisms, that is, (a) relief of craving and withdrawal [35], (b) blocking the reinforcing effects of smoking [36], (c) helping smokers smoke less [37], and (d) increasing self-efficacy [38]. If this is the case, then in principle the expense of providing extra NRT during a lapse could be justified by an increase in the number of women who successfully stop smoking altogether.

The term ‘harm reduction’ refers to a policy, strategy, or particular intervention that assumes continued use of an undesired behaviour and aspires to lower the risk of adverse consequences associated with the continuation of this addictive behaviour. While some public health advocates note that harm reduction strategies are not compatible with the ultimate goal of a tobacco-free society [39], using NRT to cut down the amount of cigarettes smoked is a strategy advocated in the general population [24] and despite stating no intention to do so, some people who use NRT like this are induced to stop smoking [40]. If NRT and concurrent smoking can be used as a tool to reduce the harm to children born to women who smoke during pregnancy and are struggling to stay abstinent, this may be an avenue worth considering, especially as qualitative work shows that some women already choose to smoke and use NRT [41].

Current advice is that, from an agreed quit date, women should stop smoking abruptly and not smoke; the ‘Not a Puff’ rule [16], however, people who use NRT and smoke need less nicotine from cigarettes so they generally smoke fewer. Non-pregnant people who used slower-acting NRT patches to cut down smoking smoked 55% fewer daily cigarettes and had 36% lower exhaled carbon monoxide (CO) levels [42]. Pregnant women who use NRT patches to cut down smoking behave similarly. Recent work has demonstrated that, compared with when smoking alone, pregnant women who used NRT and smoked consumed fewer cigarettes and exhaled less CO than when smoking alone, while cotinine concentrations (a metabolite of nicotine and used as a biomarker for nicotine exposure) remained similar between these two scenarios [43]. Although nicotine from smoking does cause temporary, dose-related increases in maternal blood pressure and heart rate when cigarettes are smoked [44], there is no convincing evidence for long term harm [45]. However, there is much evidence that tobacco smoke is harmful and there are strongly positive associations between exhaled CO levels (which can be associated with cigarette smoking [46]) and adverse neonatal outcomes [47]. As smoking alongside NRT use does not appear to increase nicotine exposure and there is no evidence that nicotine on its own is harmful in pregnancy [10,48] a case could be made for adopting a harm reduction approach when other strategies have not been successful.

At the interpersonal and individual levels, practitioners were particularly concerned about the consequences of sending out mixed messages to pregnant women. Currently SSS advise women that reducing the number of cigarettes does not equate to significantly reduced health risks and stopping smoking completely is the only way of ensuring that the unborn baby is not at risk from smoking [49]. It was felt that suggesting to women, who may be finding it difficult to quit, that they may be better trying a harm reduction approach would undermine practitioners’ current training and personal beliefs. It may also contribute to pregnant women’s confusion around NRT at a time when clear and consistent messages from health care professionals is needed [50] and increase the likelihood of women choosing and acting on information that allows them to continue to smoke when they may otherwise have successfully quit.

That practitioners found the message that concurrent use of NRT and smoking may be less harmful to the baby contentious, indicates that more research is needed in this area along with changes to practitioners’ training in light of new findings when appropriate.

At an organisational level, NRT was seen as a limited resource not to be wasted and only provided if certain conditions were met; participants’ local SSS policies were reported to dictate that NRT should only be supplied as a substitute to cigarette smoking. This may reflect the financial constraints services are under. and given most local authorities control the budget for stop smoking medications [51] it seems logical that there should be some controls on medication provision to avoid waste. In terms of harm reduction, some participants could see how offering this approach could increase the numbers of pregnant women accessing stop smoking service support, however, SSSs are currently assessed by successful quit rates [52] and so a different objective measure would be required to measure the success of a harm reduction intervention.

## 5. Strengths and Limitations

To our knowledge this is the first study that has specifically described smoking practitioners’ attitudes and beliefs around NRT use in relation to lapses or harm reduction. We have identified how organisational policies influence what NRT support can be offered and how interpersonal factors, individual beliefs and a lack of evidence in this area shape how NRT support is provided.

However, participants in these focus groups were only able to provide information about their own practice and their views are likely to have been influenced by local stop smoking service policies. While focus group sizes were appropriate to facilitate discussion, the number of groups conducted was small and findings may reflect the views of an unusual group with atypical views. Given that the provision of stop smoking services to pregnant women is complex and may be different across local authorities, the findings may not be representative of the UK as a whole. This is especially pertinent as the prevalence of smoking during pregnancy differs widely both geographically and socioeconomically. Views may also differ amongst health professionals/experts from other countries and findings may not necessarily be readily transferable to other countries. In addition, it is worth noting that we have combined the analysis of two focus groups, but while the aims of the groups were different, both were specifically asked to discuss issues concerned with NRT use in relation to lapse, relapse and harm reduction, which allowed data synthesis around these subjects possible. Also, while interaction between participants was explored, it was found to be minimal which otherwise could have added depth to the analysis.

## 6. Conclusions

Currently smoking cessation practitioners have strong reservations about recommending pregnant women to continue using NRT during all but the briefest of lapses, and they hold particularly negative views towards using NRT for harm reduction in pregnancy by those who cannot quit. Further evidence from systematic reviews and trials investigating the efficacy and safety of using NRT in lapses or for harm reduction in pregnancy, combined with qualitative findings regarding pregnant women’s views around using NRT, are required to help guide organisational policy and practitioners’ treatment decisions in these areas.

## Figures and Tables

**Table 1 ijerph-16-04791-t001:** Characteristics of focus group participants.

**Focus Group 1**				
**Participant**	**Age**	**Gender**	**Job Title**	**Experience**
1	35	F	Smoking in pregnancy specialist	>10 years
2	58	F	Research Midwife	5–10 years
3	25	F	Pregnancy lead	2–5 years
4	41	F	Senior tobacco control manager	5–10 years
5	55	F	Stop smoking in pregnancy coordinator	>10 years
6	38	F	Stop smoking service lead	>10 years
7	37	F	Stop smoking service manager & Tobacco control lead	>10 years
**Focus Group 2**				
**Participant**	**Age**	**Gender**	**Job Title**	**Experience**
1	61	F	Smoking cessation midwifery lead	>10 years
4	45	F	Stop smoking specialist trainer	>10 years
3	40	F	Specialist stop smoking advisor	2–5 years
4	48	F	Stop smoking advisor	2–5 years
5	57	F	Stop smoking advisor	2–5 years
6	56	F	Stop smoking specialist	>10 years

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
