# Peer review of "Practitioners’ Views on Nicotine Replacement Therapy in Pregnancy during Lapse and for Harm Reduction: A Qualitative Study"

_ijerph, 2019, doi:10.3390/ijerph16234791_

Round 1
Reviewer 1 Report
I think the biggest limitation of this work is the small size of the tested group. This problem should be discussed further.The authors have not explained how the results obtained outside the UK can be used.
Perhaps it would be appropriate to think of a scheme with stages of research and (qualitative) results. In its current form, work is difficult to perceive.
Author Response
Point-by-point response to Reviewer 1 comments
Dear Reviewer
Thank you for your thoughtful consideration of our manuscript. We found your comments particularly helpful, and believe our revised manuscripts, based on your suggestions, represents a significant improvement. All changes have been highlighted to aid identification. Please see the table below for a point-by-point response to your comments.
Yours sincerely
Ross Thomson
Reviewer comments |
Author reply |
I think the biggest limitation of this work is the small size of the tested group. This problem should be discussed further. |
P8 line 50-51. Text added to highlight the limitations of small number of focus groups |
The authors have not explained how the results obtained outside the UK can be used. |
P9, line 2-3. Text added to limitations section to highlight transferability of findings to other countries |
Perhaps it would be appropriate to think of a scheme with stages of research and (qualitative) results. In its current form, work is difficult to perceive. |
Unfortunately we were not confident we understood this point. However, on P9 line 15-18 we have added text to propose different stages of future work to clarify the conclusion. |
Reviewer 2 Report
Manuscript ID: ijerph-648322 Title: Practitioners’ views on Nicotine Replacement Therapy in pregnancy during lapse and for harm reduction: A qualitative study. This is a very well written and clearly presented paper exploring the views of smoking cessation practitioners with regard to the use of nicotine replacement therapy alongside smoking as a harm reduction measure in pregnant women who are unable to quit smoking. This is an important area of research which requires further in-depth research to inform policy and practice in this field. I have only minor grammatical suggestions for the authors: - Page 3, Line 33-35: the sentence is not clear. I suggest rewriting as " These questions were devised to explore how, from practitioners' and experts' perspectives, women's experiences of being guided in the use of NRT can be improved in order to encourage better use of NRT during pregnancy." - Page 4, Line 8-9: the use of the term 'concerned with;' is not clear , perhaps the sentence could start with " The included messages related to ............." ? - Page 7, Line 6: the full stop is missing. - Page 7, Line 8: 'exposed' should be 'expose'.Author Response
Point-by-point response to Reviewer 2 comments
Dear Reviewer
Thank you for your thoughtful consideration of our manuscript. We found your comments particularly helpful, and believe our revised manuscript, based on your suggestions, represents a significant improvement. All changes have been highlighted to aid identification. Please see the table below for a point-by-point response to your comments.
Yours sincerely
Ross Thomson
Reviewer comments |
Author reply |
Page 3, Line 33-35: the sentence is not clear. I suggest rewriting as " These questions were devised to explore how, from practitioners' and experts' perspectives, women's experiences of being guided in the use of NRT can be improved in order to encourage better use of NRT during pregnancy." |
Changes made and highlighted in the document |
Page 4, Line 8-9: the use of the term 'concerned with;' is not clear , perhaps the sentence could start with " The included messages related to ............." ? |
Changes made and highlighted in the document |
Page 7, Line 6: the full stop is missing. |
Changes made and highlighted in the document |
Page 7, Line 8: 'exposed' should be 'expose' |
Changes made and highlighted in the document |